# Bicuspid Aortic Valve and Sudden Cardiac Death

**DOI:** 10.3390/life15060868

**Published:** 2025-05-28

**Authors:** Cecilia Salzillo, Andrea Quaranta, Fabrizia Di Lizia, Michela Lombardo, Marco Matteo Ciccone, Vincenzo Ezio Santobuono, Enrica Macorano, Francesco Introna, Biagio Solarino, Andrea Marzullo

**Affiliations:** 1PhD Course in Public Health, Department of Experimental Medicine, University of Campania “Luigi Vanvitelli”, 80138 Naples, Italy; cecilia.salzillo@unicampania.it; 2Pathology Unit, Department of Precision and Regenerative Medicine and Ionian Area, University of Bari “Aldo Moro”, 70124 Bari, Italy; a.quaranta35@studenti.uniba.it (A.Q.); f.dilizia@studenti.uniba.it (F.D.L.); m.lombardo13@studenti.uniba.it (M.L.); 3Cardiology Unit, Interdisciplinary Department of Medicine, University of Bari “Aldo Moro”, 70124 Bari, Italy; marcomatteo.ciccone@uniba.it (M.M.C.); vincenzoezio.santobuono@uniba.it (V.E.S.); 4Legal Medicine Unit, Interdisciplinary Department of Medicine, University of Bari “Aldo Moro”, 70124 Bari, Italy; enricamacorano@gmail.com (E.M.); francesco.introna@uniba.it (F.I.); biagio.solarino@uniba.it (B.S.)

**Keywords:** bicuspid aortic valve, pathogenic variants, classification valvular, sudden cardiac death, cardiovascular complications, preventive strategies

## Abstract

Bicuspid aortic valve (BAV) is the most common congenital heart anomaly, affecting an estimated 0.5% to 0.77% of the general population. This condition occurs when the aortic valve has only two cusps instead of the usual three, disrupting normal valve function and increasing the risk of various cardiovascular diseases. Often asymptomatic in its early stages, BAV can gradually progress, leading to stenosis, valve insufficiency, and abnormalities of the ascending aorta. One particularly concerning aspect is its potential association with sudden cardiac death (SCD). The aim of this literature review is to examine the relationship between BAV and the risk of SCD, highlighting the pathogenic variants and pathophysiological mechanisms involved while emphasizing the significance of valve classification and its clinical implications. Additionally, it explores current research gaps and future directions to enhance early identification of at-risk individuals and reduce the incidence of SCD.

## 1. Introduction

Bicuspid aortic valve (BAV) is a congenital heart defect of the aortic valve, characterised by the presence of two cusps instead of three, which leads to degenerative changes in the valve and is associated with aortopathy [1,2] and sudden cardiac death (SCD) [3,4].

BAV is one of the most common congenital cardiac anomalies, with a prevalence between 0.5% and 0.77% [5], and predominantly affects males [1]. Its prevalence has increased following the advent of echocardiography [1].

BAV can be either sporadic or familial in approximately 9% of cases [4], with a probable autosomal dominant mode of transmission exhibiting incomplete penetrance and variable expressivity [6]. In accordance with the 2022 AHA/ACC guidelines, echocardiographic screening of first-degree relatives is recommended in all patients with BAV, especially when there is a family history of congenital heart disease, aortic dissection, or sudden death [7].

BAV may be associated with genetic syndromes such as Turner syndrome and Williams syndrome, with a prevalence of 30% [1], as well as Loeys–Dietz syndrome, velocardiofacial syndrome, and, occasionally, Down syndrome, Alagille syndrome, and Kabuki syndrome [5].

BAV may also be linked to congenital heart defects, including ventricular septal defects (VSD) with aortic arch obstruction in 51% of cases, adult aortic arch coarctation in 37%, isolated VSD in 20.5%, atrioventricular canals in 7.5%, Tetralogy of Fallot in 2%, and complete transposition of the great arteries in 1% [4].

BAV leads to valvular abnormalities such as incompetence, stenosis due to dystrophic calcification, and infective endocarditis [4]. In particular, aortic stenosis (AS) represents a significant and growing public health burden, particularly with aging populations. Characterized by progressive narrowing of the aortic valve, AS leads to impaired cardiac output, reduced exercise capacity, heart failure, and increased mortality [8].

BAV is also associated with aortopathies, including ascending aortic dilatation in 40–60% of cases, coarctation of the aorta in 50–75%, aortic aneurysm [1,2], aortic dissection or rupture, and SCD [4,9].

## 2. Embryology, Anatomy, and Histology

Aortic valve embryology is crucial to understanding the pathogenesis of BAV, as alterations during embryonic development directly influence the morphology and function of the valve [10,11]. The process of aortic valve formation involves the endothelial–mesenchymal transition (EndMT), a mechanism in which endothelial cells transform into mesenchymal cells and contribute to the creation of the endocardial cushions, which will subsequently allow for the rise of the valve cusps [12]. Any alteration in this process may result in incomplete separation of the cusps, leading to the formation of a bicuspid valve rather than a tricuspid one [13].

Pathogenic variants in the *NOTCH1* and *NOS3* genes have been identified as key factors in susceptibility to BAV. *NOTCH1* is essential for the regulation of cell differentiation, and its dysfunction can impair cusp fusion [14,15], while *NOS3* is involved in nitric oxide signalling, which is important for cardiovascular development [16,17].

Additionally, the extracellular matrix plays a crucial role in valve remodelling. In patients with BAV, an altered production of collagen and proteoglycans has been found, compromising the elasticity and resilience of the valve and predisposing it to early calcifications and valve dysfunction, such as stenosis and regurgitation [18].

The anatomy of the aortic valve is closely linked to its function of regulating blood flow between the left ventricle and the aorta.

Normally, the aortic valve is tricuspid (Figure 1), consisting of three thin, mobile cusps—the right, left, and non-coronary cusps—that attach to the walls of the sinuses of Valsalva.

Their microstructure comprises three main layers: the fibrous tunica or parietal layer, rich in collagen, which provides mechanical strength; the spongiosa tunica or middle layer, made of proteoglycans, which absorbs and redistributes the forces generated by blood flow; and the elastic tunica or axial or ventricular layer, composed of elastin, which allows for the flexibility needed for the cardiac cycle (Table 1).

The point of greatest connection between the cusps and the aortic wall is called the commissure, while the valve ring (annulus) represents the fibrous junction between the left ventricle and the aortic root.

In BAV, this structure is altered, with only two cusps instead of three (Figure 1). In fact, the normal tricuspid configuration is replaced by two asymmetric cusps, often resulting from the fusion of two of the three typical cusps (right, left, or non-coronary). The incomplete fusion of the cusps during embryonic development may result in the presence of an abnormal fibrous ridge called a raphe, which partially connects two cusps, allowing for the rise of different subtypes of BAV, classifiable according to the Schaefer and Sievers systems [2]. This anatomical structure affects the dynamics of blood flow, often causing turbulence that can predispose to early valve calcification and aortopathy, i.e., the progressive dilation of the ascending aorta [10,19] (Figure 2).

Another distinctive aspect of BAV anatomy concerns the diameter of the aortic annulus, which tends to be wider than that of the tricuspid aortic valve. This characteristic has significant implications for surgical and transcatheter aortic valve replacement (TAVR) procedures, as annulus width can influence the choice and implantation of prosthetic valves [20,21].

On histological examination [4], BAV typically shows signs of fibrosis and calcification affecting the valve cusps. The endothelial layer lining the valve often appears thickened and irregular, with an abnormal distribution of valve interstitial cells, which may contribute to progressive dysfunction. Additionally, the aortic wall associated with BAV frequently undergoes significant non-inflammatory degenerative changes, such as fragmentation of elastic fibres, loss of smooth muscle cells, and medial extracellular matrix alterations (Table 2).

These structural abnormalities are thought to play a key role in the development of complications like aortic dilatation and valve dysfunction over time.

## 3. Pathogenic Variants and Pathogenesis

Current single-gene pathogenic variants account for less than 10%, and the major pathogenic variants associated with BAV are *NOTCH1*, *GATA4-6*, *SMAD4* and *SMAD6*, *ROBO4*, *ACTA2*, and *FBN1* [6,22].

*NOTCH1* is the most frequently implicated genetic variant in BAV, encoding a receptor involved in endothelial–mesenchymal transition and cardiac valve development. *NOTCH1* loss-of-function mutations can accelerate aortic valve calcification in BAV [23,24,25,26]. Rare variants in other *NOTCH* pathway genes, such as *ARHGAP31*, *MAML1*, *SMARCA4*, *JARID2*, and *JAG1*, are associated with BAV and aortic coarctation [27].

*GATA4-6* encodes zinc finger transcription factors that regulate early cardiac gene expression and cardiac cell lineage differentiation [28]; specifically, common variants of *GATA4* are associated with BAV, while rare variants of *GATA4*, *GATA5*, and *GATA6* have been identified in other studies [6,29,30,31].

Variants of *SMAD4* and *SMAD6*, *TGF-β* signalling proteins, as well as rare variants of *ROBO4* expressed in endothelial cells, have been identified in BAV [6,22,32].

Mutations in *ACTA2*, encoding smooth muscle alpha-actin, are relatively rare in BAV [33]. Rare variants of *FBN1*, encoding an extracellular glycoprotein (fibrillin-1), have been found in BAV and aortic root aneurysms [6,22,34].

Table 3 summarizes the pathogenic variants associated with BAV.

Dilation of the ascending aorta is a condition commonly observed in patients with congenital aortic stenosis or insufficiency and is particularly frequent in cases of BAV. This phenomenon can be explained by two primary pathogenetic mechanisms [35,36,37,38].

The first is the genetic theory, which suggests that aortic dilation results from intrinsic alterations of the aortic wall, associated with structural defects in the extracellular matrix. These defects lead to abnormal elastin production, a protein essential for the resistance and elasticity of blood vessels. In particular, the lack of elastin compromises the ability of the aortic wall to respond adequately to pressure changes, thus promoting the progressive dilation of the vessel (Figure 3).

The second mechanism is the haemodynamic theory, which highlights the importance of blood flow within the aorta. Specifically, turbulent flow and abnormal shear stress, the tangential force exerted by blood flow on the vessel wall, are considered factors that contribute to the degeneration of the aortic wall. Abnormal haemodynamic conditions can accelerate the dilation process and weaken the vessel wall, thereby increasing the risk of complications (Figure 3).

Another important factor contributing to ascending aortic dilation is the anatomical phenotype of the aortic valve. In particular, the pattern in which the right and left cusps fuse is frequently associated with dilation of the sinus of Valsalva. This phenomenon is particularly noticeable in patients with BAV. On the other hand, the pattern in which the right and non-coronary cusps fuse is associated with a higher risk of valve dysfunction, another critical aspect of disease progression.

Dilation of the ascending aorta in patients with BAV arises from a complex interaction between genetic factors, haemodynamic mechanisms, and the anatomy of the aortic valve, which, together, promote the progressive widening of the aortic wall and increase the risk of cardiovascular complications.

## 4. Valvular Classification and Clinical Implications

BAV is characterised by considerable anatomical heterogeneity, which is why different classification systems have been developed to identify morphological variants and their clinical implications [39].

The two most used systems are those of Sievers and Schaefer.

Sievers’ classification [40] is based on the presence and number of raphes, i.e., the incomplete fusions between the cusps, and distinguishes three main types: type 0, without raphe with two completely formed cusps and two subtype; type 1, with a raphe partially joining two cusps, which is the most common variant and includes three subtypes; type 2, with two raphe and increased fusion between cusps and three subtypes (Table 4).

Schaefer’s classification [41], on the other hand, focuses on the orientation of the valve opening, dividing BAV according to the direction of the gap between the cusps: antero-posterior or lateral–lateral (Table 5).

The orientation of the cusps, as described in the Schaefer classification, can also significantly influence the dynamics of blood flow through the aortic valve. Specifically, in the A-P variant, the flow tends to remain more central, while in the L-L variant, the flow can be more eccentric. This difference may determine a different distribution of shear stress on the aortic wall, influencing the risk of dilation of the ascending aorta and the early onset of stenosis or regurgitation [39,41].

These classification systems are not merely descriptive but have significant clinical implications. For instance, Sievers type 1 is often associated with a higher risk of aortic stenosis and ascending aortic aneurysm, whereas type 0 is less predisposed to early calcification but may lead to aortic regurgitation. Despite the widespread use of these classifications, a unified consensus on their systematic clinical application remains lacking [10].

However, a recent international consensus statement has proposed a more comprehensive approach that integrates the assessment of cusp fusion, raphe calcification, symmetry, and valve annulus shape [39,42,43].

An international consensus has identified three main types of BAV, each with specific clinical and prognostic implications [39].

Fused BAV is the most frequent type, accounting for approximately 90–95% of cases. In this variant, two of the three aortic cusps are fused, resulting in the formation of two functional cusps, one of which is larger and more irregular than the other. The most common phenotype within this category is the right–left cusp fusion, occurring in 70–80% of patients with BAV. This phenotype has been associated with a higher risk of aortic stenosis with increasing age, as well as a greater predisposition to aortic root dilation and the onset of aortic regurgitation, particularly in male patients. The second-most-common phenotype, with a prevalence of 20–30%, is the right non-cusp fusion, which is more strongly associated with a faster progression of aortic stenosis and regurgitation in both children and adults. Finally, the left non-cusp fusion is the least-frequent phenotype, found in 3–6% of cases, and requires further study to fully understand its clinical and prognostic significance [39,44,45].

A second variant, two-sinus BAV, is characterised by the presence of only two aortic sinuses instead of the usual tricuspid structure with three distinct sinuses. This morphology is less common and appears to be associated with specific alterations in haemodynamic flow, potentially influencing the progression of valvular disease and the dilation of the ascending aorta [39,44,45].

The third category is partial-fusion BAV, a more subtle form of the disease in which the cusps fuse incompletely or less distinctly. This variant, sometimes referred to as frustrating forms, can be challenging to diagnose and requires more in-depth analysis with advanced imaging techniques. Although its clinical significance is not yet fully elucidated, some studies suggest that it may be associated with a slower progression of valve disease compared to fully fused BAV [39,44,45].

A detailed classification of BAV phenotypes is crucial not only for diagnosis but also for prognosis and the therapeutic management of patients. Different fusion patterns have been linked to specific risks of aortic stenosis, regurgitation, or dilation, underscoring the need for personalised follow-up and targeted therapeutic strategies. As imaging techniques and genetic research continue to advance, it will become possible to further refine our understanding of this complex condition and improve treatment options for affected patients.

Table 6 helps to understand the differences between the different types of BAV and their possible clinical implications, highlighting the importance of accurate classification for optimal patient management.

## 5. Sudden Cardiac Death, Other Cardiovascular Complications, and Preventive Strategies

BAV is one of the most prevalent congenital heart diseases in the general population, exhibiting a higher incidence in males [1,5]. In a healthy heart, the aortic valve comprises three cusps. However, in individuals with BAV, the valve consists of only two cusps, leading to significant functional and haemodynamic alterations that may compromise cardiovascular health over time [46].

Although many individuals with BAV remain asymptomatic for extended periods, the condition is strongly associated with an increased risk of severe cardiovascular complications, including SCD [4,9,47]. SCD is an unexpected fatal event, often occurring without prior warning signs, and constitutes a major cause of mortality among young athletes [48,49,50]. The presence of BAV may serve as a predisposing factor, as supported by multiple studies [51]. The underlying mechanisms of SCD in BAV include valvular dysfunction, arrhythmias, and other complications arising from structural abnormalities of the valve.

### 5.1. Haemodynamic Consequences of BAV

BAV induces substantial haemodynamic changes, particularly affecting the left ventricle. The principal pathophysiological alterations associated with BAV include:AS, characterised by narrowing of the valve orifice, is a common complication in BAV. This condition results in an increased pressure gradient across the valve, placing excessive strain on the left ventricle, which must work harder to overcome the resistance imposed by the stenotic valve. Consequently, compensatory left ventricular hypertrophy develops, potentially leading to systolic dysfunction and a heightened risk of life-threatening ventricular arrhythmias [52,53,54].

It is essential to distinguish between congenital AS associated with BAV in pediatric patients and progressive calcific stenosis that often develops later in adulthood. In the first, the valve is already stenotic at birth, often associated with other congenital anomalies such as coarctation of the aorta. In contrast, adult-onset AS in BAV typically results from accelerated degenerative calcification of the valve leaflets, which often occurs decades earlier than in tricuspid aortic valves. These two scenarios have different prognostic implications and management strategies [1,39,40,45].

Aortic regurgitation, another frequent complication, involves retrograde blood flow from the aorta into the left ventricle during diastole. This increases ventricular volume, leading to eccentric hypertrophy and eventual heart failure if left untreated [55,56,57].Aortic dilatation and dissection: BAV is often associated with progressive aortic dilatation, a significant risk factor for catastrophic events such as aortic dissection. This silent progression necessitates vigilant monitoring, as dissection can lead to sudden death if undiagnosed [58,59,60].

These haemodynamic disturbances favour an environment conducive to the development of fatal arrhythmias, a leading cause of SCD in BAV patients.

Furthermore, among the different mechanisms that contribute to SCD in patients with BAV, there is an ongoing debate on the predominant role of valvular dysfunction compared to aortic dilation. Although severe stenosis or valve insufficiency can lead to arrhythmias and heart failure, current evidence suggests that acute aortic events, particularly dissection in cases of progressive dilation, are more frequently implicated in sudden death, especially in young individuals. This reinforces the need for proactive aortic surveillance and early surgical intervention in selected cases, even in the absence of symptomatic valvulopathy [38,51].

In addition to the mechanical load imposed on the left ventricle, BAV is associated with myocardial fibrosis and alteration of ventricular geometry, particularly in the presence of long-standing stenosis or regurgitation. These structural changes may serve as a substrate for malignant ventricular arrhythmias. Some studies have identified focal scarring or late gadolinium enhancement on cardiac MRI in patients with asymptomatic BAV, suggesting that fibrotic remodelling may be a silent but significant factor contributing to arrhythmic risk [52,53,54].

### 5.2. Major Cardiovascular Complications Associated with BAV

Severe aortic stenosis is the primary complication predisposing BAV patients to SCD, particularly in athletes. The increased cardiac workload in response to stenotic obstruction raises the risk of ventricular arrhythmias and sudden death, even in previously asymptomatic individuals [61].

BAV confers an increased susceptibility to infective endocarditis, a condition in which microbial infection of the aortic valve leads to valvular destruction and systemic complications, significantly heightening the risk of SCD if untreated [62,63,64].

Individuals with BAV exhibit a higher propensity for aortic dissection due to structural fragility of the aortic wall. This catastrophic event can be fatal if prompt surgical intervention is not undertaken [65,66,67].

The exact prevalence of SCD in patients with BAV is not easily defined and varies according to the populations studied and the diagnostic criteria, but in the literature, it is generally considered relatively low in the general population with BAV, although significantly higher than in the population without BAV.

In autopsy and clinical studies, SCD in patients with BAV is estimated to occur in 0.5–1% of the BAV population, with a higher risk in young subjects, athletes, and in the presence of complications such as severe aortic stenosis, significant aortic dilation, or associated genetic syndromes [3].

Furthermore, it is necessary to distinguish between syndromic and non-syndromic BAV. In syndromic cases such as Marfan syndrome and Loeys–Dietz syndrome, the presence of systemic connective tissue diseases significantly increases the risk of aortic dissection and early vascular complications, which may predispose to SCD even in the absence of severe valvular dysfunction. In contrast, isolated BAV tends to follow a more variable course, with the risk of SCD mainly associated with mechanical complications of the valve or aorta [6].

Autopsy studies have demonstrated that BAV accounts for a significant proportion of premature cardiac-related deaths. In an analysis of 6325 hearts, 91 cases of BAV were identified, with a male predominance of 84% and an average age of death of 37 years. Among these cases, approximately 57% of deaths were attributed to valvular or aortic disease, including aortic stenosis (30%), infective endocarditis (11%), aortic dissection (9%), and aortic regurgitation (8%) [3].

### 5.3. Prevention and Management of SCD in BAV

To reduce the risk of SCD in subjects with BAV, it is essential to adopt a multidisciplinary approach that integrates several complementary strategies.

Due to the complex structural variability of BAV and its associated aortopathy, the choice of imaging modality is critical. Table 7 provides a comparative overview of the main imaging techniques used in the evaluation of BAV, highlighting their respective strengths and limitations in the evaluation of both valve morphology and aortic dimensions [2,20,43].

Early diagnosis plays a key role. In fact, transthoracic echocardiography is the main diagnostic tool for assessing the morphology and functionality of the aortic valve. However, in some clinical situations, more advanced imaging techniques, such as cardiac magnetic resonance imaging (CMR) or computed tomography (CT), may be necessary to obtain a more in-depth and precise assessment.

Regular monitoring of valve function and aortic dimensions allows early detection of any signs of disease progression and timely intervention, avoiding serious complications, and is based on a structured follow-up protocol based on disease severity. In patients with mild valvular disease and stable aortic dimensions, echocardiographic follow-up is recommended every 2–3 years. In case of moderate valvular disease or borderline aortic dilation (40–45 mm), annual follow-up is suggested. In the case of severe valvular disease or rapidly expanding aortas (>5 mm/year), follow-up should be performed every 6–12 months. Timing may also be influenced by the patient’s age, blood pressure control, and presence of symptoms [7].

Although the mechanical consequences of BAV, such as aortic stenosis and dilation, are recognized as contributing factors to SCD, emerging research suggests that certain genetic variants may play a direct role. Mutations in genes such as *NOTCH1*, *SMAD6*, *ACTA2*, and *FBN1* not only influence valve morphology and aortic wall integrity but can also influence myocardial tissue composition and electrical stability. This raises the hypothesis that, in some patients, SCD may occur even in the absence of advanced hemodynamic lesions due to a genetically determined arrhythmogenic substrate [6,22,23,24,25,26,33,34].

Clinical management of asymptomatic patients with normal valve function simply requires constant surveillance. However, in the presence of significant aortic stenosis or regurgitation, surgical intervention, such as valve replacement, may be necessary. Current clinical guidelines recommend routine surveillance every 6–12 months in patients with severe but still asymptomatic aortic stenosis. However, new data are changing the therapeutic landscape: the recently published EARLY TAVR study [68] demonstrated that treatment with transcatheter valve replacement (TAVI) in asymptomatic patients with severe aortic stenosis reduced the risk of a composite endpoint, including death, stroke, or unplanned cardiovascular hospitalizations by 50%, compared to the watchful waiting strategy, over a follow-up of almost 4 years (HR 0.50; 95% CI 0.40–0.63). In light of these results, the FDA recently approved the extension of the indication for TAVI to asymptomatic patients with severe aortic stenosis [69]. This opens new perspectives for a more proactive management of this population, potentially significantly improving clinical outcomes. Due to the strong association between BAV and aortic pathology, it is essential to carefully monitor aortic dimensions and consider prophylactic surgery in cases of significant dilation, even in the absence of obvious symptoms.

It is important to emphasize that the recommended cut-off values for surgery on the root or ascending aorta differ between patients with tricuspid aortic valve and BAV. According to the ESC/EACTS 2021 guidelines for the management of valvular heart disease (Vahanian et al., 2022) [70], in subjects with BAV, surgery is generally indicated at a diameter ≥ 55 mm but can be considered already at 50 mm in the presence of risk factors such as rapid aortic growth, family history of dissection, or associated coarctation.

Furthermore, TAVR outcomes in patients with BAV are significantly influenced by valve morphology. In particular, the presence of a calcified raphe and extensive calcification of the valve leaflets is associated with an increased risk of procedural complications and medium-term mortality. Patients who exhibit both of these characteristics are the ones who most frequently experience complications during the procedure [71].

TAVR in patients with BAV has also been linked to a higher incidence of procedural complications compared to the tricuspid valve. However, this difference was more marked with first-generation devices. More recent data indicate that new-generation prostheses have improved outcomes also in patients with BAV [72,73].

Despite exclusion from major randomized trials, patients with BAV are increasingly being treated with TAVI. Large observational analyses and meta-analyses suggest that the results, in terms of safety and efficacy, are comparable to those observed in patients with tricuspid valve, even in low surgical risk groups [74].

TAVR in BAV remains technically challenging, with a relatively higher risk of paravalvular regurgitation (PVR). In Sievers type 1 BAV, moderate or severe PVR occurs in ~4% of cases even with the latest generation valves and is associated with higher rates of major adverse events (MAEs) during follow-up [75].

Finally, it is important to promote education and awareness, both among patients and healthcare professionals. Greater knowledge of the risk factors and early signs of SCD can facilitate timely diagnosis and more effective life-saving interventions.

Although current guidelines emphasize the importance of regular clinical and imaging monitoring, validated risk scores or specifically developed quantitative models to predict SCD in patients with BAV are still lacking. Some studies have proposed risk stratification based on valve morphology, aortic diameter thresholds, or the presence of pathogenic genetic variants. However, their predictive value is still being studied and has not yet been translated into routine clinical practice. Further research is needed to define robust and individualized tools to better identify high-risk patients [3,55,67].

Intervention strategies can significantly influence the risk of SCD in patients with BAV. Surgical aortic valve replacement and prophylactic aortic root surgery, when indicated, have been shown to reduce the risk of dissection and SCD. Medical therapy, such as beta-blockers or angiotensin II receptor antagonists, may help stabilize aortic size and reduce wall stress, particularly in genetically predisposed individuals. Although medical treatment alone cannot eliminate the risk of SCD, it helps to modulate the disease and mitigate the risk when combined with regular imaging and timely surgical intervention [65,66,67].

## 6. Conclusions and Future Perspectives

BAV represents one of the most frequent congenital cardiac anomalies, with important clinical implications related to the risk of stenosis, regurgitation, aortopathy, and SCD. Although diagnostics and follow-up have been greatly improved thanks to advanced imaging and genetic characterization, significant knowledge gaps still exist, especially regarding therapeutic management.

In particular, the increasing use of TAVR in patients with BAV, despite exclusion from many randomized trials, highlights the urgency of dedicated prospective studies. It is essential to evaluate the efficacy and safety of TAVR in this heterogeneous population, identifying morphological and clinical predictors of procedural success and long-term outcomes.

Only through targeted studies will it be possible to optimize the treatment of BAV and improve risk stratification. Indeed, in the not-too-distant future, a multidisciplinary approach combining advanced technologies, genetics, and personalized medicine could really make a difference in reducing the risk of SCD in patients with BAV, significantly improving their quality and life expectancy.

## Figures and Tables

**Figure 1 life-15-00868-f001:**
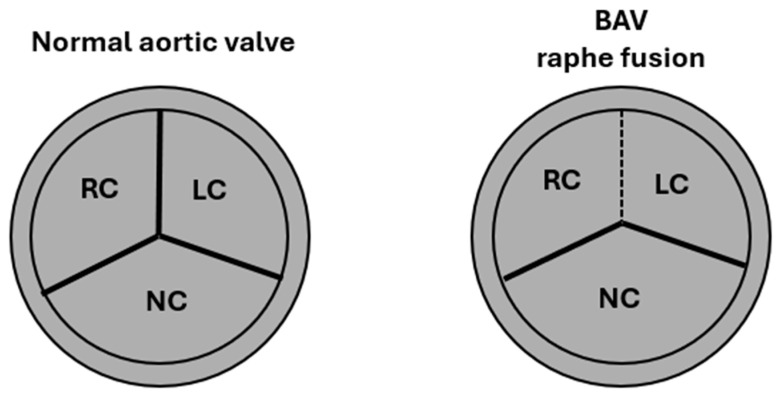
Normal aortic valve with three cusps and bicuspid aortic valve with fusion of the raphe. RC, right cusp; LC, left cusp; NC, noncoronary cusp.

**Figure 2 life-15-00868-f002:**
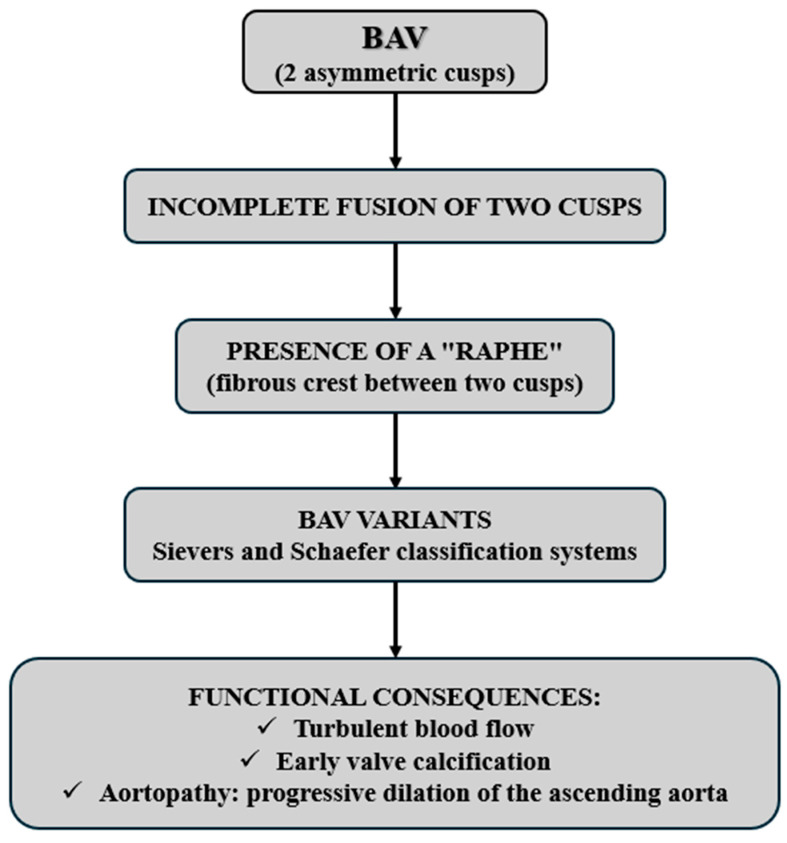
Bicuspid aortic valve anatomy and functional consequences.

**Figure 3 life-15-00868-f003:**
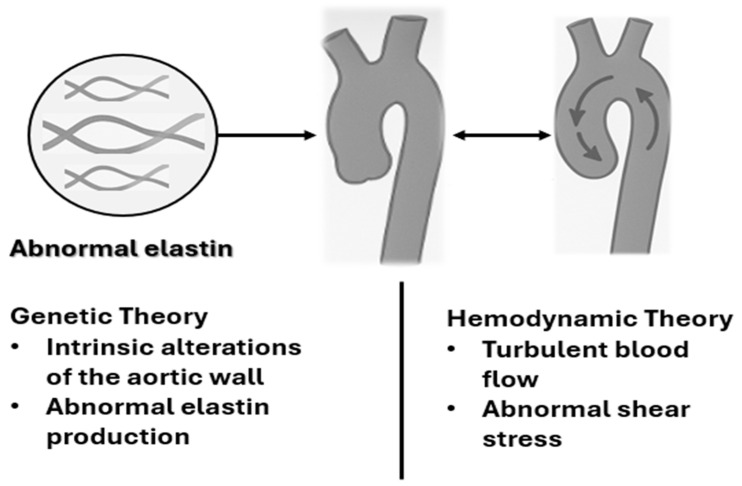
Pathogenetic mechanisms of aortic dilation.

**Table 1 life-15-00868-t001:** Structure and function of the aortic valve.

Tunica/Layers	Structure	Function
Fibrous tunica (parietal layer)	Collagen	It is stretched in diastole to maintain flap adhesion
Tunica spongiosa (middle layer)	Proteoglycans	Absorbs and distributes forces and cyclic movements of the valve
Elastic tunica (axial or ventricular layer)	Elastin	It relaxes in diastole and contracts in systole

**Table 2 life-15-00868-t002:** BAV histological features.

Histology	Description
Fibrosis and calcification	Accumulation of fibrotic tissue and calcium deposits on valve cusps
Endothelial thickening	Irregular endothelium with anomalous distribution of interstitial cells
Elastic fiber fragmentation	Disintegration of the elastic structure of the aortic wall
Smooth muscle cell loss	Reduction of muscle cells in the aortic media, contributing to wall weakness
Extracellular matrix alterations	Pathologic remodeling of the extracellular matrix with accumulation of proteoglycans

**Table 3 life-15-00868-t003:** Current genes and pathogenic variants associated with BAV.

Pathogenic Variants	Function	Role in BAV
NOTCH1	Receptor involved in endothelial-mesenchymal transition and heart valve development	Variant most frequently associated with BAV, and loss-of-function mutations accelerate aortic valve calcification
ARHGAP31, MAML1, SMARCA4, JARID2, JAG1	NOTCH pathway genes	Rare variants associated with BAV and aortic coarctation
GATA4-6	Transcription factors with zinc finger domain	Regulate early cardiac gene expression and cardiac cell differentiation, common variants of GATA4 associated with BAV, while rare variants of GATA4, GATA5, and GATA6 have been identified in other studies
SMAD4, SMAD6	TGF-β signalling proteins	Variants identified in BAV
ROBO4	Expressed in endothelial cells	Rare variants identified in BAV
ACTA2	Encodes smooth muscle alpha-actin	Relatively rare mutations in BAV
FBN1	Encodes extra-cellular glycoprotein fibrillin 1	Rare variants found in BAV and aortic root aneurysms

**Table 4 life-15-00868-t004:** Sievers’ classification of BAV (based on raphe).

Type	Description	Subtype	Main Features
0	No raphe, two completely separated cusps	A-P Lateral	More symmetrical structure and less predisposition to calcifications
1	A raphe, partial fusion of two cusps	L-R, R-NC, L-NC	The most common variant and often associated with aortic stenosis and dilation of the ascending aorta
2	Two raphes, extensive fusion	L-R/R-NC R-NC/NC-L NC-L/L-R	Greater alteration of blood flow and associated with more abnormal valves

A-P, Anterior-Posterior; L, Left; R, Right; NC, Non-Coronary.

**Table 5 life-15-00868-t005:** Schaefer’s classification of BAV (based on the orientation of the cusps).

Type	Description	Main Features
A-P	The cusps are oriented anteroposteriorly	Most common, often associated with aortic stenosis
L-L	The cusps are laterally oriented	Less common, it can affect hemodynamics differently

A-P, Anterior-Posterior; L-L, Lateral–Lateral.

**Table 6 life-15-00868-t006:** BAV types and their main phenotypes, characteristics, and clinical implications.

Type	Main Phenotypes	Prevalence	Anatomical Features	Clinical Implications
Fused BAV	Right–left cusp fusion	70–80%	Fusion between left and right cusp	Increased risk of aortic stenosis, aortic root dilation, aortic regurgitation (especially in males)
	Right non-cusp fusion	20–30%	Fusion between right and non-coronary cusp	Faster progression of aortic stenosis and regurgitation
	Left non-cusp fusion	3–6%	Fusion between left and non-coronary cusp	Rare phenotype, further studies needed
Two-sinus BAV	No specific phenotype	Rare	Presence of only two aortic sinuses	Alterations in hemodynamic flow, potential dilation of the ascending aorta
Partial-fusion BAV	No specific phenotype	Undetermined	Incomplete or less evident fusion of the cusps	Slower progression of valvular disease, difficult to diagnose with traditional imaging

**Table 7 life-15-00868-t007:** Comparison of imaging techniques for BAV phenotyping and aortic assessment.

Imaging Modality	Main Advantages	Limitations	Application in BAV Phenotypes
Transthoracic Echocardiography (TTE)	Non-invasive, widely available, useful for assessing valve function.	Limited accuracy for distal aortic measurements or in patients with poor acoustic window.	First approach for screening, useful for valve follow-up.
Transoesophageal Echocardiography (TEE)	Higher resolution than TTE, excellent visualization of the aortic root and valve.	Invasive, not optimal for distal ascending aorta.	Useful for detailed morphological evaluation of BAV.
Computed Tomography (CT)	High spatial resolution, excellent for aorta evaluation.	Exposure to radiation and iodinated contrast medium.	Optimal for accurate aortic measurements, especially presurgical.
Cardiovascular Magnetic Resonance Imaging (MRI)	No radiation exposure, good for aortic measurements and ventricular function.	Limitations in patients with metallic devices or claustrophobia.	Useful in young patients or for long-term follow-up of aortic dilation.

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
