# Peer review of "Bicuspid Aortic Valve and Sudden Cardiac Death"

_life, 2025, doi:10.3390/life15060868_

Round 1
Reviewer 1 Report
Comments and Suggestions for Authors
The submitted manuscript presents a comprehensive review of the current knowledge on bicuspid aortic valve disease, with a particular focus on its association with sudden cardiac death, integrating insights from embryology, genetics, histopathology, and clinical management.
The paper is logically structured and flows well from one section to another. Technical terms are well defined and explained. Tables summarize information effectively.
While the manuscript is well written, covering most important and relevant topics, it should be clear that the review does not introduce novel hypotheses or particularly groundbreaking perspectives.
The authors however have an opportunity to enhance the manuscript and make it more up-to date, based on the following suggestions:
1. The management of aortic stenosis (AS) should be more extensively described. In section “5.3. Prevention and management of SCD in BAV – Clinical Management” the authors should mention the recent approval of one of the TAVI systems for asymptomatic patients with AS. Suggested wording: While current clinical guidelines recommend routine surveillance every 6 to 12 months in patients with asymptomatic severe aortic stenosis, the recently published EARLY TAVR trial (Généreux P, et. al.; EARLY TAVR Trial Investigators. Transcatheter Aortic-Valve Replacement for Asymptomatic Severe Aortic Stenosis. N Engl J Med. 2025 Jan 16;392(3):217-227. doi: 10.1056/NEJMoa2405880.) demonstrated that treating patients who had severe aortic stenosis but no symptoms with TAVI halved the risk of a composite endpoint of death, stroke, or unplanned cardiovascular hospitalizations over nearly 4 years of follow-up when compared with watchful waiting (HR 0.50; 95% CI 0.40-0.63). Based on this, the recent FDA approval, led to the indication expansion of TAVI to asymptomatic patients with severe aortic stenosis. Ref: https://ir.edwards.com/news/news-details/2025/Edwards-TAVR-Receives-FDA-Approval-for-Patients-With-Asymptomatic-Severe-Aortic-Stenosis/default.aspx (accessed on 10.05.2025)
2. The introduction should benefit from a more general overview about the burden of the disease. Authors may use a similar suggested wording: Aortic stenosis (AS) represents a significant and growing public health burden, particularly with aging populations. Characterized by progressive narrowing of the aortic valve, AS leads to impaired cardiac output, reduced exercise capacity, heart failure, and increased mortality. Please use the reference: Roleder T, et. al. Trends in diagnosis and treatment of aortic stenosis in the years 2006-2016 according to the SILCARD registry. Pol Arch Intern Med. 2018 Dec 21;128(12):739-745. doi: 10.20452/pamw.4352.
3. A few sentences are repetitive or redundant, especially in the introduction and conclusion. Please double check again for a better flow and clarity.
Comments on the Quality of English Language3There are some grammatical inconsistencies or stylistic errors such as “classification valvular” should be “valvular classification” – Section 4 headline, line 164
Author Response
Dear Reviewer1,
Thank you for your comments that improve the manuscript and as suggested:
Comments 1: The management of aortic stenosis (AS) should be more extensively described. In section “5.3. Prevention and management of SCD in BAV – Clinical Management” the authors should mention the recent approval of one of the TAVI systems for asymptomatic patients with AS. Suggested wording: While current clinical guidelines recommend routine surveillance every 6 to 12 months in patients with asymptomatic severe aortic stenosis, the recently published EARLY TAVR trial (Généreux P, et. al.; EARLY TAVR Trial Investigators. Transcatheter Aortic-Valve Replacement for Asymptomatic Severe Aortic Stenosis. N Engl J Med. 2025 Jan 16;392(3):217-227. doi: 10.1056/NEJMoa2405880.) demonstrated that treating patients who had severe aortic stenosis but no symptoms with TAVI halved the risk of a composite endpoint of death, stroke, or unplanned cardiovascular hospitalizations over nearly 4 years of follow-up when compared with watchful waiting (HR 0.50; 95% CI 0.40-0.63). Based on this, the recent FDA approval, led to the indication expansion of TAVI to asymptomatic patients with severe aortic stenosis. Ref: https://ir.edwards.com/news/news-details/2025/Edwards-TAVR-Receives-FDA-Approval-for-Patients-With-Asymptomatic-Severe-Aortic-Stenosis/default.aspx (accessed on 10.05.2025)
Response 1: We have expanded the section "5.3. Prevention and management of SCD in BAV – Clinical Management" (377-392, red highlight) and inserted the two references (71 and 72: red highlight)
Comments 2: The introduction should benefit from a more general overview about the burden of the disease. Authors may use a similar suggested wording: Aortic stenosis (AS) represents a significant and growing public health burden, particularly with aging populations. Characterized by progressive narrowing of the aortic valve, AS leads to impaired cardiac output, reduced exercise capacity, heart failure, and increased mortality. Please use the reference: Roleder T, et. al. Trends in diagnosis and treatment of aortic stenosis in the years 2006-2016 according to the SILCARD registry. Pol Arch Intern Med. 2018 Dec 21;128(12):739-745. doi: 10.20452/pamw.4352.
Response 2: We have inserted the general overview in the introduction (49-53, red highlight) and inserted the citation (8 and red highlight)
Comments 3:
-A few sentences are repetitive or redundant, especially in the introduction and conclusion. Please double check again for a better flow and clarity.
-There are some grammatical inconsistencies or stylistic errors such as “classification valvular” should be “valvular classification” – Section 4 headline, line 164
Response 3:
-We have checked the entire text eliminating redundancies. We have rewritten the conclusions in a more concise and focused way, emphasizing the urgent need for prospective studies dedicated to the evaluation of TAVR in patients with BAV (highlighted in light blue).
-We have corrected "Valvular classification" (paragraph 4 and red highlight). We have also had the English corrected by an expert.
Thank you and kind regards.
Reviewer 2 Report
Comments and Suggestions for Authors
The authors provide a comprehensive and quite well referenced review of the he relationship between Bicuspid Aortic Valve (BAV) and sudden cardiac death (SCD). It presents embryological, genetic, anatomical, pathological, and clinical aspects with sufficient depth and appropriate referencing. As they prepare a revised version of this review, I have several overall arching and some specific requirements and recommendations. Individually and as a collective, these are meant to clarify the text, improve the Tables and provide further referencing and documentation in support of key background material.
- General Comments: It is apparent that very significant goal directed literature review and document planning has been done. Nonetheless, there are several important areas and sections that may be improved. Examples include:
- a) There is no discussion of risk scores or quantitative models that predict SCD in BAV patients such as risk thresholds based on valve morphology, aortic diameter, genetic mutations.
- b) I also have a question for the authors: Whether aortic dilation or valve dysfunction contributes more to SCD?
- c) you may also add a focused section or table comparing the diagnostic value of TTE, TEE, CT, and MRI for BAV phenotypes and aortic measurements.
- Specific Edits: When rewriting your manuscript please replace or improve each of the following:
There are minor grammatical inconsistencies:
"classification valvular" should be "valvular classification".
“anomalous fibrous ridge, called a raphe, which partially connects two cusps" should be: "an abnormal fibrous ridge called a raphe, which partially connects two cusps"
Comments on the Quality of English LanguageThere are minor grammatical inconsistencies:
"classification valvular" should be "valvular classification".
“anomalous fibrous ridge, called a raphe, which partially connects two cusps" should be: "an abnormal fibrous ridge called a raphe, which partially connects two cusps"
Author Response
Dear Reviewer2,
Thank you for your comments that improve the manuscript and as suggested:
Comments 1: General Comments: It is apparent that very significant goal directed literature review and document planning has been done. Nonetheless, there are several important areas and sections that may be improved. Examples include:
2.a) There is no discussion of risk scores or quantitative models that predict SCD in BAV patients such as risk thresholds based on valve morphology, aortic diameter, genetic mutations.
3.b) I also have a question for the authors: Whether aortic dilation or valve dysfunction contributes more to SCD?
4.c) you may also add a focused section or table comparing the diagnostic value of TTE, TEE, CT, and MRI for BAV phenotypes and aortic measurements.
Response 1:
2a) We have added a paragraph in section "5.3. Prevention and management of SCD in BAV" discussing the lack of validated quantitative models for the prediction of SCD in patients with BAV (419-425, highlighted in yellow) and inserted the references.
3b) We have integrated in section "5.1. Haemodynamic consequences of BAV" a paragraph that compares the role of valvular dysfunction and aortic dilatation in the genesis of SCD (294- 301, highlighted in yellow) and inserted the references.
4c) We have inserted a paragraph (346-350, highlighted in yellow) and a new table (Table 7) in section 5.3 (352), comparing the main imaging techniques used in the evaluation of BAV and associated aortopathy.
Comments 2: Specific Edits: When rewriting your manuscript please replace or improve each of the following:
There are minor grammatical inconsistencies:
"classification valvular" should be "valvular classification".
“anomalous fibrous ridge, called a raphe, which partially connects two cusps" should be: "an abnormal fibrous ridge called a raphe, which partially connects two cusps"
Comments on the Quality of English Language
There are minor grammatical inconsistencies:
"classification valvular" should be "valvular classification".
“anomalous fibrous ridge, called a raphe, which partially connects two cusps" should be: "an abnormal fibrous ridge called a raphe, which partially connects two cusps"
Response 2:
-We have corrected "Valvular classification" (177, paragraph 4 and red highlight).
-We have corrected "an abnormal fibrous ridge called a raphe, which partially connects two cusps" (98, paragraph 2 and highlighted in yellow).
We have also had the English corrected by an expert.
Thank you and kind regards.
Reviewer 3 Report
Comments and Suggestions for Authors
The topic of this review is interesting, but uncommon in patients with BAV. Probably for this reason, the SCD is covered in only approximately 20% of the article. In my opinion it should be expanded.
My suggestions for the issues to be discussed:
- The prevalence of SCD in BAV and the difference between syndromic versus non-syndromic BAV
- SCD could be also related to the genetic variants or occurs as a consequence of BAV’s well known complications?
- More comments related to the mechanisms of SCD
- To highlight the distinction between bicuspid aortic valve (BAV) associated with congenital aortic stenosis (AS) in pediatric patients and BAV in which AS develops later due to progressive valve calcification
- What means regular monitoring in patients with BAV and how should it be done?
- When should family screening be done?
- When CT and MRI are usefull to follow-up patients with BAV beyond the echocardiography?
- Medical or surgical treatment can influence the risk of SCD in patients with BAV?
Other comments
- There is a discrepancy in the reported prevalence of BAV between the abstract and the main article, and the prevalence value is repeated twice within the article.
- A diagram illustrating the anatomy of the bicuspid aortic valve (BAV) would be useful along with the description in the text.
- Abbreviations must be explained: RL and RN pattern ; TAVR.
- I recommend revising the titles of the following tables:
Table 3 Current genes and pathogenic variants associated with BAV
Table 4. Sievers' classification of BAV (based on raphe)
Table 5. Schaefer’s classification of BAV (based on the orientation of the cusps).
- Table 5: „... it can affect hemodynamics differently”. Could you explain how the hemodynamics are affected differently?
English can be improved
Author Response
Dear Reviewer3,
Thank you for your comments that improve the manuscript and as suggested:
Comments 1: The prevalence of SCD in BAV and the difference between syndromic versus non-syndromic BAV
Response 1: We have expanded section "5.2. Major cardiovascular complications associated with BAV" to include an estimate of the prevalence of SCD in patients with BAV and have also introduced a commentary on the increased vulnerability in cases of syndromic BAV (321-335, highlighted in green).
Comments 2: SCD could be also related to the genetic variants or occurs as a consequence of BAV’s well known complications?
Response 2: We added in section "5.3. Prevention and management of SCD" that, in addition to the known hemodynamic complications, some studies suggest that genetic variants may predispose to greater myocardial or aortic instability, potentially contributing to arrhythmic events and SCD, even independently of the morphological severity of the valve (369-376, highlighted in green).
Comments 3: More comments related to the mechanisms of SCD
Response 3: We have integrated in section "5.1. Haemodynamic consequences of BAV" a paragraph that compares the role of valvular dysfunction and aortic dilatation in the genesis of SCD (294-301, highlighted in yellow). We have added in the same section further details on arrhythmic mechanisms, in particular, left ventricular hypertrophy secondary to stenosis, myocardial fibrosis and left ventricular remodelling may represent substrates for malignant ventricular arrhythmias (302-308, highlighted green).
Comments 4: To highlight the distinction between bicuspid aortic valve (BAV) associated with congenital aortic stenosis (AS) in pediatric patients and BAV in which AS develops later due to progressive valve calcification
Response 4: We have added a paragraph in "section 5.1" clarifying the distinction between the two clinical scenarios (276-283, highlighted in green).
Comments 5: What means regular monitoring in patients with BAV and how should it be done?
Response 5: We have inserted in "section 5.3" a paragraph to clarify that “regular monitoring” in patients with BAV refers to echocardiographic checks, depending on the presence and severity of aortic stenosis and dilation (360-368, highlighted in purple).
Comments 6: When should family screening be done?
Response 6: We have better specified in the introduction that, in accordance with the AHA/ACC 2022 guidelines, echocardiographic screening of first-degree relatives is recommended in patients with BAV, particularly if there is a family history of congenital heart disease, aortic disease or sudden death (36-39, highlighted in green).
Comments 7: When CT and MRI are usefull to follow-up patients with BAV beyond the echocardiography?
Response 7: We have inserted a paragraph (346-350, highlighted in yellow) and a new table (352, Table 7) in section 5.3, comparing the main imaging techniques used in the follow-up of BAV and associated aortopathy.
Comments 8: Medical or surgical treatment can influence the risk of SCD in patients with BAV?
Response 8: We added at the end of "section 5.3" that early surgical treatment, such as valve replacement or prophylactic surgery of the ascending aorta, is the only intervention currently recognized to significantly reduce the risk of SCD. Medical treatment has a more indirect preventive role, in containing the progression of valvular and aortic disease. (426-433, highlighted in green)
Comments 9: There is a discrepancy in the reported prevalence of BAV between the abstract and the main article, and the prevalence value is repeated twice within the article.
Response 9: We corrected the prevalence in the abstract (13, highlighted in green) and eliminated the repetition of the prevalence percentage at the beginning of section 5 (254-255, highlighted in green).
Comments 10: A diagram illustrating the anatomy of the bicuspid aortic valve (BAV) would be useful along with the description in the text.
Response 10: We added figure 2.
Comments 11: Abbreviations must be explained: RL and RN pattern ; TAVR.
Response 11: We have removed the acronyms RL and RN as it is explained in detail (highlighted in green, 167 and 169). We explained TAVR (highlighted in green, 109).
Comments 12: I recommend revising the titles of the following tables:
Table 3 Current genes and pathogenic variants associated with BAV
Table 4. Sievers' classification of BAV (based on raphe)
Table 5. Schaefer’s classification of BAV (based on the orientation of the cusps).
Response 12: We have changed the table titles (3, 4, 5) as indicated. (highlighted in green)
Comments 13: Table 5: „... it can affect hemodynamics differently”. Could you explain how the hemodynamics are affected differently?
Response 13: We have inserted a paragraph in the text (198-203, below Table 5 and highlighted in green) to clarify how hemodynamics may be influenced differently in relation to the orientation of the cusps in the Schaefer classification.
Comments 14: English can be improved
Response 14: We have also had the English corrected by an expert.
Thank you and kind regards.
Reviewer 4 Report
Comments and Suggestions for Authors
The review is timely and relevant, particularly given the increasing diagnosis of bicuspid aortic valve (BAV) facilitated by advances in imaging techniques. The content is clearly structured and easy to follow. However, certain aspects would benefit from more in-depth discussion or emphasis.
1) Concering TAVR in BAV patients: it is crucial to highlight the earlier onset of associated conditions—such as aortic ectasia, aortic stenosis, and aortic regurgitation—in patients with bicuspid aortic valve (BAV) compared to the general population. Historically, this led to the referral of BAV patients for surgical aortic valve replacement (SAVR). However, with the recent expansion of TAVR indications in the guidelines, careful assessment of valve morphology has become essential to identify those who may achieve better outcomes with transcatheter treatment. The following points should be emphasized given their relevance to the management of BAV disease.
-It should be emphasized that the recommended cut-off values for ascending aortic diameter differ between tricuspid and bicuspid aortic valves in both American and European guidelines (Vahanian A, Beyersdorf F, Praz F, et al. ESC/EACTS Scientific Document Group. 2021 ESC/EACTS Guidelines for the management of valvular heart disease. Eur Heart J. 2022 Feb 12;43(7):561-632. doi: 10.1093/eurheartj/ehab395. Erratum in: Eur Heart J. 2022 Feb 18;: PMID: 34453165.)
-Outcomes of TAVR in patients with bicuspid aortic stenosis are influenced by valve morphology: the presence of a calcified raphe and extensive leaflet calcification are associated with a higher risk of procedural complications and midterm mortality. In particular, patients with both features experience procedural complications more frequently. (Yoon SH, Kim WK, Dhoble A, et al. Bicuspid Aortic Valve Stenosis Transcatheter Aortic Valve Replacement Registry Investigators. Bicuspid Aortic Valve Morphology and Outcomes After Transcatheter Aortic Valve Replacement. J Am Coll Cardiol. 2020 Sep 1;76(9):1018-1030. doi: 10.1016/j.jacc.2020.07.005. PMID: 32854836.)
-TAVR in patients with BAV aortic stenosis was linked to a higher incidence of procedural complications compared to tricuspid AS, though this difference was primarily seen with early-generation devices and not with the newer-generation (Yoon SH, Bleiziffer S, De Backer O, et al Outcomes in Transcatheter Aortic Valve Replacement for Bicuspid Versus Tricuspid Aortic Valve Stenosis. J Am Coll Cardiol. 2017 May 30;69(21):2579-2589. doi: 10.1016/j.jacc.2017.03.017. Epub 2017 Mar 18. PMID: 28330793; Halim SA, Edwards FH, Dai D, et al Outcomes of Transcatheter Aortic Valve Replacement in Patients With Bicuspid Aortic Valve Disease: A Report From the Society of Thoracic Surgeons/American College of Cardiology Transcatheter Valve Therapy Registry. 2020 Mar 31;141(13):1071-1079. doi 10.1161/CIRCULATIONAHA.119.040333. Epub 2020 Feb 26. PMID: 32098500)
-Despite being excluded from major trials, BAV patients are commonly treated with TAVI, with large studies showing safety and outcomes comparable to TAV patients, even in low-risk groups. (Saeed Al-Asad K, Martinez Salazar A, Radwan Y et al Transcatheter Aortic Valve Replacement in Bicuspid Versus Tricuspid Aortic Valve Stenosis: Meta-Analysis and Systemic Review. Am J Cardiol. 2023 Sep 15;203:105-112. doi: 10.1016/j.amjcard.2023.06.120. Epub 2023 Jul 22. PMID: 37487404.
-TAVR in BAV stenosis remains technically challenging, with an elevated risk of paravalvular regurgitation (PVR). In Sievers type 1 BAV, moderate to severe PVR occurs in ~4% of cases with current-generation valves and is linked to higher rates of major adverse events (MAEs) during follow-up. (Zito A, Buono A, Scotti A, et al Predictors, and Outcomes of Paravalvular Regurgitation After TAVR in Sievers Type 1 Bicuspid Aortic Valves. JACC Cardiovasc Interv. 2024 Jul 22;17(14):1652-1663. doi: 10.1016/j.jcin.2024.05.002. Epub 2024 May 14. PMID: 38749449.)
2) The conclusions should be more concise and should emphasize the need for dedicated studies evaluating TAVR in BAV patients, especially in light of the expanding use of this treatment modality.
Author Response
Dear Reviewer4,
Thank you for your comments that improve the manuscript and as suggested:
Comments 1: The review is timely and relevant, particularly given the increasing diagnosis of bicuspid aortic valve (BAV) facilitated by advances in imaging techniques. The content is clearly structured and easy to follow. However, certain aspects would benefit from more in-depth discussion or emphasis.
1) Concering TAVR in BAV patients: it is crucial to highlight the earlier onset of associated conditions—such as aortic ectasia, aortic stenosis, and aortic regurgitation—in patients with bicuspid aortic valve (BAV) compared to the general population. Historically, this led to the referral of BAV patients for surgical aortic valve replacement (SAVR). However, with the recent expansion of TAVR indications in the guidelines, careful assessment of valve morphology has become essential to identify those who may achieve better outcomes with transcatheter treatment. The following points should be emphasized given their relevance to the management of BAV disease.
1-It should be emphasized that the recommended cut-off values for ascending aortic diameter differ between tricuspid and bicuspid aortic valves in both American and European guidelines (Vahanian A, Beyersdorf F, Praz F, et al. ESC/EACTS Scientific Document Group. 2021 ESC/EACTS Guidelines for the management of valvular heart disease. Eur Heart J. 2022 Feb 12;43(7):561-632. doi: 10.1093/eurheartj/ehab395. Erratum in: Eur Heart J. 2022 Feb 18;: PMID: 34453165.)
2-Outcomes of TAVR in patients with bicuspid aortic stenosis are influenced by valve morphology: the presence of a calcified raphe and extensive leaflet calcification are associated with a higher risk of procedural complications and midterm mortality. In particular, patients with both features experience procedural complications more frequently. (Yoon SH, Kim WK, Dhoble A, et al. Bicuspid Aortic Valve Stenosis Transcatheter Aortic Valve Replacement Registry Investigators. Bicuspid Aortic Valve Morphology and Outcomes After Transcatheter Aortic Valve Replacement. J Am Coll Cardiol. 2020 Sep 1;76(9):1018-1030. doi: 10.1016/j.jacc.2020.07.005. PMID: 32854836.)
3-TAVR in patients with BAV aortic stenosis was linked to a higher incidence of procedural complications compared to tricuspid AS, though this difference was primarily seen with early-generation devices and not with the newer-generation (Yoon SH, Bleiziffer S, De Backer O, et al Outcomes in Transcatheter Aortic Valve Replacement for Bicuspid Versus Tricuspid Aortic Valve Stenosis. J Am Coll Cardiol. 2017 May 30;69(21):2579-2589. doi: 10.1016/j.jacc.2017.03.017. Epub 2017 Mar 18. PMID: 28330793; Halim SA, Edwards FH, Dai D, et al Outcomes of Transcatheter Aortic Valve Replacement in Patients With Bicuspid Aortic Valve Disease: A Report From the Society of Thoracic Surgeons/American College of Cardiology Transcatheter Valve Therapy Registry. 2020 Mar 31;141(13):1071-1079. doi 10.1161/CIRCULATIONAHA.119.040333. Epub 2020 Feb 26. PMID: 32098500)
4-Despite being excluded from major trials, BAV patients are commonly treated with TAVI, with large studies showing safety and outcomes comparable to TAV patients, even in low-risk groups. (Saeed Al-Asad K, Martinez Salazar A, Radwan Y et al Transcatheter Aortic Valve Replacement in Bicuspid Versus Tricuspid Aortic Valve Stenosis: Meta-Analysis and Systemic Review. Am J Cardiol. 2023 Sep 15;203:105-112. doi: 10.1016/j.amjcard.2023.06.120. Epub 2023 Jul 22. PMID: 37487404.
5-TAVR in BAV stenosis remains technically challenging, with an elevated risk of paravalvular regurgitation (PVR). In Sievers type 1 BAV, moderate to severe PVR occurs in ~4% of cases with current-generation valves and is linked to higher rates of major adverse events (MAEs) during follow-up. (Zito A, Buono A, Scotti A, et al Predictors, and Outcomes of Paravalvular Regurgitation After TAVR in Sievers Type 1 Bicuspid Aortic Valves. JACC Cardiovasc Interv. 2024 Jul 22;17(14):1652-1663. doi: 10.1016/j.jcin.2024.05.002. Epub 2024 May 14. PMID: 38749449.)
Response 1: We have inserted the paragraphs with the citations (73-78) in the manuscript section 5.3 (393-415, highlighted in light blue).
Comments 2: The conclusions should be more concise and should emphasize the need for dedicated studies evaluating TAVR in BAV patients, especially in light of the expanding use of this treatment modality.
Response 2: We have rewritten the conclusions in a more concise and focused way, emphasizing the urgent need for prospective studies dedicated to the evaluation of TAVR in patients with BAV (highlighted in light blue).
We have also had the English corrected by an expert.
Thank you and kind regards.
Round 2
Reviewer 1 Report
Comments and Suggestions for Authors
The manuscript improved significantly. I have no further comments
Reviewer 2 Report
Comments and Suggestions for Authors
The authors improved the manuscript. I have no other comments.
Reviewer 3 Report
Comments and Suggestions for Authors
All my comments and suggestions were fulfilled. Thank you!